# Hypothesis Testing the Circuit Hypothesis in LLMs

**Claudia Shi**[*1]         **Nicolas Beltran-Velez**[* 1]         **Achille Nazaret**[* 1]
**Carolina Zheng**[* 1]         **Adrià Garriga-Alonso**[3]         **Andrew Jesson**[1]
**Maggie Makar**[2]                                 **David M. Blei**[1]

[1]Department of Computer Science, Columbia University, New York, USA
[2]Computer Science and Engineering, University of Michigan, Ann Arbor, USA
[3]FAR AI, USA

## Abstract

Large language models (LLMs) demonstrate surprising capabilities, but we do not understand how they are implemented. One hypothesis suggests that these capabilities are primarily executed by small subnetworks within the LLM, known as circuits. But how can we evaluate this hypothesis? In this paper, we formalize a set of criteria that a circuit is hypothesized to meet and develop a suite of hypothesis tests to evaluate how well circuits satisfy them. The criteria focus on the extent to which the LLM's behavior is preserved, the degree of localization of this behavior, and whether the circuit is minimal. We apply these tests to six circuits described in the research literature. We find that synthetic circuits – circuits that are hard-coded in the model – align with the idealized properties. Circuits discovered in Transformer models satisfy the criteria to varying degrees. To facilitate future empirical studies of circuits, we created the *circuitry* package, a wrapper around the *TransformerLens* library, which abstracts away lower-level manipulations of hooks and activations. The software is available at `https://github.com/blei-lab/circuitry`.

## 1 Introduction

The field of mechanistic interpretability aims to explain the inner workings of large language models (LLMs) through reverse engineering. One promising direction is to identify "circuits" that correspond to different tasks. Examples include circuits that perform context repetition [Olsson et al., 2022], identify indirect objects [Wang et al., 2023], and complete docstrings [Heimersheim and Janiak, 2023].

Such research is motivated by the circuit hypothesis, which posits that LLMs implement their capabilities via small subnetworks within the model. If the circuit hypothesis holds, it would be scientifically interesting and practically useful. For example, it could lead to valuable insights about the emergence of properties such as in-context learning [Olsson et al., 2022] and grokking during training [Stander et al., 2023, Nanda et al., 2023b]. Moreover, identifying these circuits could aid in explaining model performance and controlling model output, such as improving truthfulness.

In this work, we empirically study the circuit hypothesis to assess its validity in practice. We begin by defining the ideal properties of circuits, which we posit to be: 1. *Mechanism Preservation:* The performance of an idealized circuit should match that of the original model. 2. *Mechanism*

---

[*]Equal contribution.

*Localization:* Removing the circuit should eliminate the model's ability to perform the associated task. 3. *Minimality:* A circuit should not contain any redundant edges.

We translate these properties into testable hypotheses. Some of these hypotheses depend on the strict validity of the idealized circuit hypothesis, while others are more flexible, allowing us to quantify the extent to which discovered circuits align with the ideal properties.

We apply these tests to six circuits described in the literature that each correspond to a different task: two synthetic, hard-coded circuits and four discovered in Transformer models. These circuits have also been used to benchmark automatic circuit discovery algorithms [Conmy et al., 2023, Syed et al., 2023].

We find that the synthetic circuits align well with the idealized properties and our hypotheses while the discovered circuits do not strictly adhere to the idealized properties. Nevertheless, these circuits are far from being random subnetworks within the model. Among the discovered circuits, the induction circuit [Olsson et al., 2022] passes two of the three idealized tests. The Docstring circuit [Heimersheim and Janiak, 2023] passes the minimality test. Furthermore, the empirical results indicate that these circuits can be significantly improved, bringing them closer to idealized circuits. For example, for two of them, removing 20% of the edges had little impact on their ability to approximate the model.

The contributions of this paper are: 1. A suite of formal and testable hypotheses derived from the circuit hypothesis. 2. A set of statistical procedures and software to perform each test. 3. An empirical study of existing circuits and their alignment to the circuit hypothesis.

## 1.1 Related work

This research fits in the broader field of mechanistic interpretability. We provide a brief overview of related work here and a more comprehensive discussion in § A.

Olah et al. [2020] introduced the concept of a circuit. Subsequently, various circuits have been proposed, particularly in vision models [Mu and Andreas, 2020, Cammarata et al., 2021, Schubert et al., 2021] and language models [Olsson et al., 2022, Wang et al., 2023, Hanna et al., 2023, Lieberum et al., 2023]. The literature has been especially effective in explaining small Transformers that perform algorithmic tasks [Nanda et al., 2023a, Heimersheim and Janiak, 2023, Zhong et al., 2023, Quirke et al., 2023, Stander et al., 2023].

This work builds on the growing effort around evaluating the quality of interpretability results [Doshi-Velez and Kim, 2017, Casper et al., 2023, Mills et al., 2023, Hase et al., 2024, Jacovi and Goldberg, 2020, Geiger et al., 2021, Chan et al., 2022, Wang et al., 2023, Schwettmann et al., 2023, Lindner et al., 2024, Friedman et al., 2024, Variengien and Winsor, 2023]. It is closely related to the works of Wang et al. [2023] and Conmy et al. [2023]. Wang et al. [2023] introduce three criteria – faithfulness, minimality, and completeness – to evaluate the Indirect Object Identification circuit. Faithfulness serves as a metric, while minimality and completeness involve searching the space of circuits. Our idealized criteria are similar in spirit to Wang et al. [2023], but the specific tests differ. A key distinction is our adoption of a hypothesis testing framework, where none of our tests require searching the space of circuits. Conmy et al. [2023] develops an automatic circuit discovery algorithm and assess the quality of circuits by measuring edge classification quality against a set of benchmark circuits.

## 2 Mechanistic Interpretability and LLMs

In this section, we define the necessary ingredients for mechanistic interpretability in LLMs.[1]

### 2.1 LLMs as computation graphs

A Transformer-based LLM is a neural network that takes in a sequence of input tokens and produces a sequence of logits over possible output tokens. We define it as a function $M : \mathcal{X} \to \mathcal{O}$, where $\mathcal{X} = \{(x^1, \dots, x^L) | \ x^\ell \in V, \ L \in \mathbb{Z}_{\geq 1}\}$ is the space of sequences of tokens, $V$ is the space of possible tokens, called the vocabulary, and $\mathcal{O} = \{(o^1, \dots, o^L) | \ o^\ell \in \mathbb{R}^{|V|}, \ L \in \mathbb{Z}_{\geq 1}\}$ is the space of sequences of logits over the vocabulary.

---

[1]For details about the Transformer architecture, see Elhage et al. [2021] for an excellent overview.

The function $M$ is computed by a sequence of smaller operations that compose to form a **computational graph**. A computational graph is a directed acyclic graph $\mathcal{G} = (\mathcal{V}, \mathcal{E})$, where $\mathcal{V}$ is the set of nodes and $\mathcal{E}$ is the set of edges. Each node $v \in \mathcal{V}$ represents an operation with one or more inputs and a single output. Each edge $(u, v) \in \mathcal{E}$ denotes that the output of node $u$ is used as the input to node $v$. We recursively define the output of node $v$ as $a_v = v(a_v^{\text{in}})$, where $a_v^{\text{in}} = \{a_u \mid u \in \mathcal{V}, (u, v) \in \mathcal{E}\}$ are the inputs to $v$. We denote the number of inputs to $v$ as $d_v$.

We can use different levels of granularity to define the nodes of a computational graph, each leading to different types of interpretability. Following Elhage et al. [2021], we define the nodes of the computational graph of an LLM to be attention heads and MLP layers. The edges correspond to the residual connections between them. We also include input nodes corresponding to the embeddings of the input tokens and output nodes corresponding to the logits.

## 2.2 Tasks: measuring the performance of a model

To measure whether a particular model performs a specific function, we define a *task*, $\tau$, as a tuple $\tau = (\mathcal{D}, s)$ of a dataset $\mathcal{D} = \{(x_i, y_i)\}_{i=1}^n$ and a score $s : \mathcal{O} \times \mathcal{Y} \to \mathbb{R}$. The dataset $\mathcal{D}$ contains pairs of inputs $x_i \in \mathcal{X}$ and output $y_i \in \mathcal{Y}$. The score maps a sequence of logits, such as the output of the model $M(x_i)$, and the ground truth information $y_i$ to a real number indicating the performance of the model's output on that particular example: a higher score indicates better performance.

**Example 1** (Greater-Than). *An example task is the* greater-than *operation [Hanna et al., 2023], where we evaluate whether the model can perform this task as it would appear in natural language. The dataset $\mathcal{D}$ contains inputs from $x_i =$ "The* `noun` *lasted from the year* `XXYY` *to the year* `XX`*" where* `noun` *is an event, e.g "war", * `XX` *is a century, e.g.* `16`*, and* `YY` *is a specific year in the century. The score function is the difference in assigned probabilities between the years smaller than $y_i =$* `YY` *and the years greater than or equal to $y_i$. The implied task is to predict the next token* `YY'` *as any year greater than* `YY` *so as to respect chronological order.*

A **circuit** is a subgraph $C = (\mathcal{V}_C, \mathcal{E}_C)$ of the computational graph $\mathcal{G}$. It includes the input and output nodes and a subset of edges, $\mathcal{E}_C$, that connect the input to the output. We let $\mathcal{C}$ denote the space of all circuits. Fig. 1 depicts one such circuit in a simplified computational graph of a two-layer attention-only Transformer. Given a circuit, we define its **complement** $\bar{C}$ to be the subgraph of $\mathcal{G}$ that includes all edges not in $C$ and their corresponding nodes.

## 2.3 Circuits of an LLM

A circuit specifies a valid subgraph, but it is not sufficient to specify a runnable model. Recall that a node $v$ in the circuit is a function with a collection of inputs $a_u$ corresponding to each $(u, v)$ present in $\mathcal{E}$. If the edge $(u, v)$ is removed, then what input $a_u$ should be provided to node $v$?

One solution, called **activation patching**, is to replace all inputs $a_u$ with an alternative value $a_u^*$, one for each edge $(u, v) \in \mathcal{E}$ that is absent from the circuit.

There are various ways to choose a value for $a_u^*$. Two common approaches are zero ablation, which sets $a_u^*$ to 0 [Olsson et al., 2022], and Symmetric Token Replacement (STR) patching [Chan et al., 2022, Geiger et al., 2024, Zhang and Nanda, 2024]. STR sets $a_u^*$ differently for each input $x_i$ and proceeds as: First, create a corrupted input $x_i^c$, which should be like $x_i$ but with key tokens changed to semantically similar ones. For example, in the greater-than task with input $x_i =$ "The war lasted from the year 1973 to the year 19", we might replace it with $x_i^c =$ "The war lasted from the year 1901 to the year 19". The meaning is preserved but the $\geq 73$ constraint is removed. Then, run the model on $x_i^c$ and cache all the activations $a_u^*$. Finally, run the circuit on $x_i$, replacing the input $a_u$ of $v$ with the cached $a_u^*$ for all edges $(u, v) \in \mathcal{E} \backslash \mathcal{E}_C$ iteratively until reaching at the output node.

We use the notation $C(x)$ to denote the output of the circuit $C$ on the input $x$, where the ablation scheme is implicit. When we compute the output of the complement of the circuit, namely $\bar{C}(x)$, we say that we *knock out* the circuit $C$ from the model $M$.

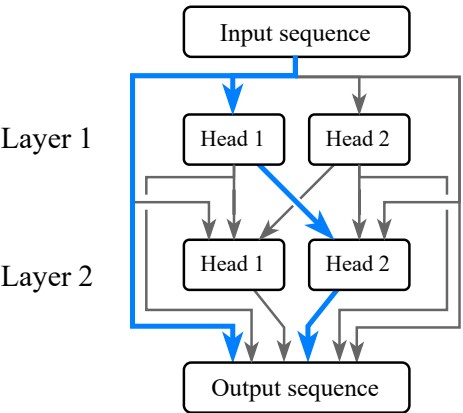

Figure 1: Simplified computational graph of a two-layer LLM with two attention heads (without MLPs). Nodes in each layer connect to all nodes in the next layer via residual connections. A highlighted arbitrary circuit is shown in blue. In a detailed graph, each incoming edge to an attention head splits into three: query, key, and value.

## 2.4 Evaluation metric: faithfulness

Given a circuit $C(x)$ and a task $\tau = (\mathcal{D}, s)$, we can use the score function to evaluate how well the circuit performs the task. However, in mechanistic interpretability, the goal is often to evaluate whether the circuit replicates the behavior of the model, which is known as faithfulness.

We define a faithfulness metric, $F : \mathcal{C} \times \mathcal{C} \times \mathcal{X} \times \mathcal{Y} \to \mathbb{R}$, that maps two circuits and an example in $\mathcal{D}$ to a real number measuring the similarity in behavior of these two circuits with respect to this particular example. We then define the **faithfulness** of circuit $C$ to model $M$ on task $\tau$ as

$$F_\tau(M, C) := \mathbb{E}_{(X,Y) \sim \mathcal{D}} \left[ F(M, C, X, Y) \right]. \tag{1}$$

We call $F_\tau(M, C)$ the *faithfulness score* of circuit $C$. For example, a faithfulness metric could be the $l^k$ norm between the score of the model and the score of the circuit,

$$F(M, C, x, y) = |s(M(x), y) - s(C(x), y)|^k,$$

with $k \in \{1, 2\}$. However, $F$ can be more general and non-symmetric, such as the KL divergence between the logits of $M$ and $C$ [Conmy et al., 2023]. Following convention, a lower value for $F_\tau(M, C)$ means the circuit is more faithful to the model.

## 3 Hypothesis Testing on Circuits

In this section, we develop three tests that formalize the following idealized criteria for a circuit: 1. *Mechanism Preservation*: The circuit should approximate the original model's performance on the task. 2. *Mechanism Localization*: The circuit should include all information critical to the task's execution. 3. *Minimality*: The circuit should be as small as possible.

In § 3.1, we discuss three idealized but stringent hypotheses implied by these criteria and develop tests for each. Then, in § 3.2, we develop two flexible tests for *Mechanism Localization* and *Mechanism Preservation*, allowing users to design the null hypotheses and determine to what extent a circuit aligns with the idealized properties.

Standard hypothesis testing has 5 components:

1. A variable of interest, $Z^*$, e.g., the faithfulness of the candidate circuit.
2. A reference distribution $\mathcal{P}_Z$ over other $Z$ that we wish to compare $Z^*$ to, along with $n$ samples $(z_i)_{i=1}^n$ from it, e.g., the faithfulness of $n$ randomly sampled circuits.
3. A null hypothesis $H_0$, which relates $Z^*$ and $\mathcal{P}_Z$ and which we assume holds true. E.g., $H_0$: "the candidate circuit is less faithful than $90\%$ of random circuits from $\mathcal{P}_Z$."

4. A real-valued statistic $t((z_i)_{i=1}^n)$ computed from the $n$ samples, with known distribution when $H_0$ holds, e.g, the number of times the candidate circuit is less faithful than a random circuit is a binomial variable with success probability $90\%$ if $H_0$ holds.

5. A confidence level $1 - \alpha$ and a rejection region $R_\alpha \subset \mathbb{R}$ such that if $H_0$ is true, then the test statistic falls in $R_\alpha$ with probability less than $\alpha$. If we observe that $t$ does fall in $R_\alpha$, we conclude that $H_0$ is false and we reject $H_0$. We will be correct $1 - \alpha$ of the time when $H_0$ is true.

Finally, defining a rejection region for each $\alpha$, the $p$-value is the smallest $\alpha$ such that $t((z_i)_{i=1}^n) \in R_\alpha$ [Young and Smith, 2005]. The smaller the $p$-value, the stronger the evidence against $H_0$.

To perform a hypothesis test, we specify the five components above, obtain the samples $z_1, \ldots, z_n$, compute the test statistic $t(z_1, \ldots, z_n)$ with the associated $p$-value, and reject the null hypothesis with confidence $1 - \alpha$ if $t(z_1, \ldots z_n) \in R_\alpha$.

### 3.1 Idealized tests

We develop three tests, *Equivalence*, *Independence*, and *Minimality*, which are direct implications of the idealized criteria. These tests are designed to be stringent: if a circuit passes them, it provides strong evidence that the circuit aligns with the idealized criteria.

We assume we have a model $M$, a task $\tau = (\mathcal{D}, s)$ with a score function $s$, and a faithfulness metric $F$. We are then given a candidate circuit $C^*$ to evaluate.

**Equivalence.** Intuitively, if $C^*$ is a good approximation of the original model $M$, then $C^*$ should perform as well as $M$ on any random task input. Hence, the difference in task performance between $M$ and $C^*$ should be indistinguishable from chance. We formalize this intuition with an equivalence test: *the circuit and the original model should have the same chance of outperforming each other.*

We write the difference in the task performance between the candidate circuit and the original model on one task datapoint $(x, y)$ as $\Delta(x, y) = s(C^*(x); y) - s(M(x); y)$, and let the null hypothesis be

$$H_0 : \left| \mathbb{P}_{(X,Y)\sim\mathcal{D}} \left( \Delta\left(X, Y\right) > 0 \right) - \frac{1}{2} \right| < \epsilon, \tag{2}$$

where $\epsilon > 0$ specifies a tolerance level for the difference in performance.

To test this hypothesis, we use a nonparametric test designed specifically for null hypotheses like $H_0$. The test statistic is the number of times $C^*$ and $M$ outperform each other. We provide a detailed description of the test in § B.1.

Since $H_0$ is in the idealized direction, if we reject the null, we claim with confidence $1 - \alpha$,

> **Non-Equivalence:** $C^*$ *and* $M$ *are unlikely to be equivalent on random task data.*

**Independence.** If a circuit is solely responsible for the operations relevant to a task, then knocking it out would render the complement circuit unable to perform the task. An implication is that the performance of the complement circuit is independent of the original model on the task.

To formalize this claim, we define the null hypothesis as

$$H_0 : s(\overline{C^*}(X); Y) \perp\!\!\!\perp s(M(X); Y), \tag{3}$$

where the randomness is over $X$ and $Y$.

To test this hypothesis, we use a permutation test. Specifically, we measure the independence between the performance of the complement circuit and the performance of the original model by using the Hilbert Schmidt Independence Criterion (HSIC) [Gretton et al., 2007], a nonparametric measure of independence. We provide a formal definition of HSIC and describe the test in § B.3.

If the null is rejected, it implies that the complement circuit and the original model's performances are not independent. We claim with confidence $1 - \alpha$,

> **Non-Independence:** *Knocking out the candidate circuit does not remove all the information relevant to the task that is present in the original model.*

**Minimality.** For minimality, we ask whether the circuit contains unnecessary edges, which are defined to be edges which when removed do not significantly change the circuit's performance.

Formally, we define the change induced by removing an edge $e \in \mathcal{E}_C$ from a circuit $C$ as

$$\delta(e, C) = \mathbb{E}_{(x,y) \sim \mathcal{D}} \left| s(C(x), y) - s(C_{-e}(x), y) \right|, \tag{4}$$

where $C_{-e} = (\mathcal{V}, \mathcal{E}_C \backslash \{e\})$.

We are interested in knowing whether for some specific edge $e^* \in \mathcal{E}_C$ the value $\delta(e^*, C^*)$ is significant. The problem now becomes how to define the reference distribution against which to compare $\delta(e^*, C^*)$. Ideally, we would like to form the distribution $\delta(e, C^*)$ induced by unnecessary edges $e$. But we do not know which $e$ in $C^*$ are unnecessary (finding them is precisely our goal).

To address this problem, we augment $C^*$ to create "inflated" circuits. An inflated circuit $C^I$ of $C^*$ is obtained by adding a random path to $C^*$ that introduces at least one new edge. Our assumption is that the randomly added path is unnecessary to the circuit performance, and so removing one of the added edges and studying the change in performance will provide our reference distribution.

We define $(C^I, e^I) \sim \mathcal{R}^I$ such that $C^I$ is a random inflated circuit obtained with the above procedure, and $e^I$ is an edge sampled uniformly at random over the novel edges $\mathcal{E}_{C^I} \backslash \mathcal{E}_{C^*}$. We then compare $\delta(e^*, C^*)$, the change induced by removing edge $e^*$ from $C^*$, against the distribution of the random variable $\delta(e^I, C^I)$, the change induced by removing what is assumed to be an irrelevant edge from an inflated version of $C^*$. A graphical illustration of this procedure is in Fig. 4.

We define the null hypothesis as

$$H_0 : \mathbb{P}_{C^I, e^I \sim \mathcal{R}^I} (\delta(e^*, C^*) > \delta(e^I, C^I)) > q^*, \tag{5}$$

where $q^*$ is a prespecified quantile.

The null hypothesis states that removing the edge $e^* \in \mathcal{E}_C$ induces a significant change in the circuit score compared to removing the random edge in the inflated circuit. If we reject $H_0$, we have found an unnecessary edge. We use a tail test in Algorithm 1 to compute the $p$-value.

If we perform the test on multiple different edges in the circuit, we need to correct for multiple hypothesis testing. To do so we use the Bonferroni correction, which is a conservative correction that controls the family-wise error rate [Dunn, 1961]. If we test $m$ edges in the circuit, the corrected significance level is $\alpha/m$.

If we test against multiple edges and after Bonferroni correction there is at least one edge for which the null hypothesis is rejected, then we claim with confidence $1 - \alpha$,

> **Non-Minimality:** *The circuit has unnecessary edges.*

## 3.2 Flexible tests

§ 3.1 presents stringent tests that align with the idealized versions of circuits. Passing any of these tests is a notable achievement for any circuit. Here, we consider two flexible ways of testing mechanism preservation (*sufficiency*) and mechanism location (*partial necessity*).

Instead of comparing the candidate circuit to the original model, we compare $C^*$ against random circuits drawn from a reference distribution. Different definitions of the reference distribution modulate the difficulty of the tests. We demonstrate that by varying the definition of the reference distribution, we can determine the extent to which the circuit aligns with the idealized criteria.

**Sufficiency.** For the sufficiency test, we ask whether the candidate circuit is particularly faithful to the original model compared to a random circuit from a reference distribution.

The variable of interest is the faithfulness of $C^*$ to $M$, $Z^* = F_\tau(M, C^*)$. We define the reference distribution $\mathcal{P}_Z$ as the distribution of $Z = F_\tau(M, C^r)$ induced by sampling random circuits $C^r$ from a chosen distribution $\mathcal{R}$. The null hypothesis is

$$H_0 : \mathbb{P}_{C^r \sim \mathcal{R}} (F_\tau(M, C^*) < F_\tau(M, C^r)) \leq q^*, \tag{6}$$

where $q^*$ is a prespecified quantile.

| Test | Tracr-P | Tracr-R | **Induction** | **IOI** | **G-T** | **DS** |
|---|---|---|---|---|---|---|
| Equivalence | ✓ | ✓ | × | × | × | × |
| Independence | ✓ | ✓ | ✓ | × | × | × |
| Minimality | ✓ | ✓ | ✓ | × | × | ✓ |

Table 1: Hypothesis testing results for six circuits using the three idealized tests. A (✓) indicates the null hypothesis is rejected, while a (×) indicates it is retained. The gray shaded boxes denote synthetic circuits, which align with our hypothesized behavior. For the equivalence test, $\epsilon = 0.1$.

The advantage of a null hypothesis like Eq. 6 is that we can change the reference distribution $\mathcal{R}$ and quantile $q^*$ to capture to what degree we test the circuit hypothesis. For example, an easier (but important) version of the test is to have the reference distribution be over all circuits of the same size as $C^*$. This test will verify that the candidate circuit is not simply a draw from the distribution of random circuits, ensuring that it is better than at least a fraction $q^*$ of random circuits.

Moreover, we can modulate the difficulty and the implied conclusions of the test by changing the size of the random circuits relative to $C^*$ and/or the target quantile $q^*$. If our distribution of random circuits produces a fraction $\eta$ of circuits that are supersets of $C^*$ (which we expect to be comparable to $C^*$), we can set $q^* = 1 - \eta$, an upper bound for the test's stringency.

The test statistic for Eq. 6 is the proportion of times $C^*$ is more faithful than $C_i^r$ for $n$ circuits $C_i^r$ sampled from $\mathcal{R}$, $t(C_1^r, \ldots, C_n^r) = \sum_{i=1}^n \frac{\mathbb{1}\{F_\tau(M, C^*) < F_\tau(M, C_i^r)\}}{n}$. Under $H_0$, the test statistic follows a binomial distribution, and we compute the associated $p$-value using a one-sided binomial test. This procedure is described in more detail in Algorithm 1 of § B.

If the $p$-value is less than the significance level $\alpha$ for a quantile $q^*$, we claim with confidence $1 - \alpha$,

> **Sufficiency:** *The probability that $C^*$ is more faithful to $M$ than $C^r$ is at least $q^*$.*

**Partial necessity.**    If the candidate circuit is responsible for solving a task in the model, then removing it will impair the model's ability to perform the task. However, this impairment may not be so severe as to make the model entirely independent of the complement circuit's output as tested in the independence test.

Instead, we define *partial necessity*: compared to removing a random reference circuit, removing the candidate circuit significantly reduces the model's faithfulness. The null hypothesis is

$$H_0 : \mathbb{P}_{C^r \sim R}(C^* \text{ is worse to knock out than } C^r) \leq q^*, \tag{7}$$

where $q^* \in (0, 1)$ is a user-chosen parameter and where "$C^*$ is worse to knock out than $C^r$" is short-hand for $F_\tau(M, \overline{C^*}) > F_\tau(M, \overline{C^r})$.

Similar to the sufficiency test, this hypothesis test is highly flexible in its design. An easier version involves using a reference distribution over circuits from the complement $\overline{C^*}$ distribution. This allows us to determine whether the edges in the candidate circuit are particularly important for task performance compared to a random circuit. Another approach is to define the reference distribution by sampling from the original model $M$, enabling us to assess whether the significance of a knockdown effect could have occurred by chance.

The test statistic is the proportion of times that $\overline{C^*}$ is less faithful than $\overline{C^r}$. Similar to the sufficiency test, we apply a binomial test to get the $p$-value. If $H_0$ is rejected, we claim with confidence $1 - \alpha$,

> **Partial necessity:** *The probability that knocking out $C^*$ damages the faithfulness to $M$ more than knocking out a random reference circuit is at least $q^*$.*

## 4    Empirical Studies

We apply hypothesis tests to six benchmark circuits from the literature: two synthetic and four manually discovered. The synthetic circuits align with the idealized properties, validating our criteria.

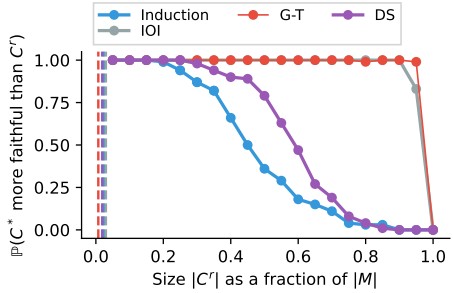 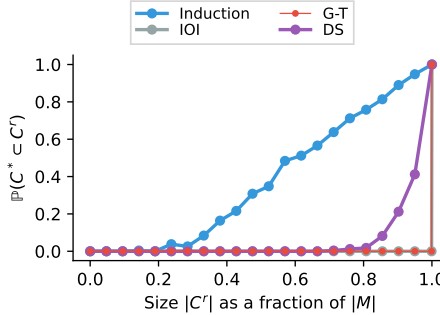

Figure 2: Left: The relative faithfulness of the candidate circuit compared to a random circuit from the reference distribution of varying sizes (x-axis). Dotted vertical lines indicate the actual size of the circuits. Right: The probability that a random circuit contains the canonical circuit.

While the discovered circuits align with the hypotheses to varying degrees, the tests help assess their quality and analyze how well each circuit aligns with the idealized criteria.

## 4.1 Experimental setup

We use the experiment configuration from ACDC [Conmy et al., 2023] for all tasks and circuits and perform the ablations using TransformerLens [Nanda and Bloom, 2022]. Below, we briefly describe each task, with detailed explanations in § D. We omit the greater-than (G-T) task as it was detailed in § 2. Both IOI and greater-than use GPT-2 small, while the other tasks use various small Transformers.

**Indirect Object Identification (IOI, Wang et al. 2023)**: The goal is to predict the indirect object in a sentence containing two entities. For example, given the sequence "When Mary and John went for a walk, John gave an apple to", the task is to predict the token " Mary". The score function is logit(" Mary") − logit(" John").

**Induction (Olsson et al. 2022)**: The objective is to predict B after a sequence of the form AB...A. For example, given the sequence "Vernon Dursley and Petunia Durs", the goal is to predict the token "ley" [Elhage et al., 2021]. The score function for this task is the log probability assigned to the correct token.

**Docstring (DS, Heimersheim and Janiak 2023)**: The objective is to predict the next variable name in a Python docstring. The score function is the logit difference between the correct answer and the most positive logit over the set of alternative arguments.

**Tracr (Lindner et al. 2024)**: For `tracr-r`, the goal is to reverse an input sequence. For `tracr-p`, the goal is to compute the proportion of x tokens in the input. The score function is the $\ell^2$ distance between the correct and predicted output. Both of these tasks have "ground truth" circuits, as the Transformers are compiled RASP programs Weiss et al. [2021], hence we call them synthetic circuits.

**Experiment details.** To construct the reference distributions $\mathcal{R}$ of random circuits for the different tests, we sample paths in $M$ (or $\overline{C^*}$) from the input nodes (embeddings) to the output node (logits) using a random walk. For the sufficiency and partial necessity tests, we start from an empty circuit and augment it with the sampled paths until it has at least $k$ edges, where $k$ is a number we vary in our experiments. For the minimality test, we inflate the circuit by adding one randomly sampled path, and we then randomly choose an edge in the added path to knock out. We draw 100 random circuits to form the reference distribution for the sufficiency and partial necessity tests. For minimality, we draw 10,000 random edges for G-T and IOI and 1000 random edges for the other circuits. In all experiments, we use Eq. 1 with $\ell^2$ norm as the faithfulness metric. We set $q^*$ to be 0.9 and $\alpha$ to be 0.05.

## 4.2 Results

Below we report and analyze key findings across tests. Additional results are reported in § E.

**Idealized tests.** Table 1 presents the results for the six circuits across the three idealized hypothesis tests. The synthetic circuits (highlighted in grey) align well with our hypotheses, supporting the validity of these tests. Among the discovered circuits, alignment with the idealized hypotheses varies:

the Induction circuit passes both the minimality and independence tests, while the Docstring circuit pass the minimality test. However, none of the other discovered circuits satisfy the idealized tests. Notably, both the Induction and Docstring circuits were identified in toy Transformer models and are relatively small in size. § E.

**Sufficiency test.** We apply the sufficiency test to study the extent to which existing circuits align with the circuit hypothesis. As noted in § 3.2, we can adjust the reference distribution to vary the test's stringency. Fig. 2 (left) illustrates the relative faithfulness of the candidate circuits compared to different reference circuit distributions.

The IOI and G-T circuits are significantly more faithful than random circuits at $90\%$ of the original model's size, while the DS and Induction circuits outperform random circuits at $30\%$ size. These results suggest their faithfulness is not due to random chance.

If the circuit hypothesis holds, we can expect the probability a randomly sampled circuit is as faithful as the candidate circuit to be equal to the probability the random circuit contains the candidate circuit. In Fig. 2 (right), we illustrate this probability under our sampling algorithm. We observe that the curve on the left is similar to the inverse of the right. Notably, while the Induction and DS circuits appear similar in Fig. 2 (left), they differ in Fig. 2 (right). The difference suggests that the Induction circuit is more closely aligned with the idealized properties compared to the DS circuit.

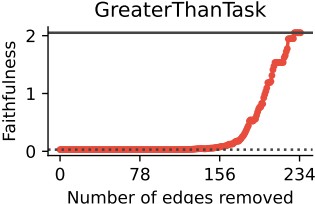 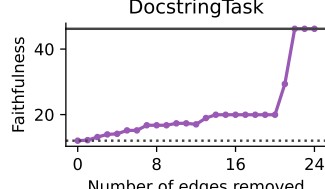 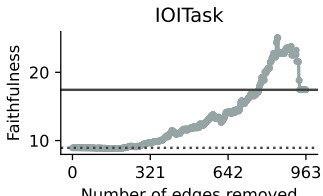

Figure 3: The faithfulness of the circuit as we gradually knock down more edges from the canonical circuit. Edges are removed in order of their minimality score, starting with the least minimal. The dotted line shows the canonical circuit's faithfulness, and the solid line shows an empty circuit's faithfulness. Removing a few minimal edges does not significantly affect faithfulness.

**Partial necessity test.** We now analyze the knockdown effect of the candidate circuit. Similar to the sufficiency test, we can define different reference distributions that reflect different underlying hypotheses. Table 2 reports the results of the hypothesis tests under two reference distributions.

We observe that when the reference circuit is drawn from the complement $\overline{C^*}$, the knockdown effect for the candidate circuits is significant across tasks. This suggests that edges in the candidate circuit play a more significant role in task performance than edges in the complement circuit. However, when compared against reference circuits drawn from the model $M$, we find that knocking down the candidate circuit does not always have a more significant effect than knocking down a random circuit of the same size.

We increase the size of the random circuits to further investigate when knocking down a random circuit has a more significant effect than knocking down the candidate circuit. Surprisingly, we observed that for G-T and IOI, knocking down the complete model has less impact than knocking down the candidate circuit. This pattern appears with the STR ablation scheme but is absent with zero-ablation. Specifically, with zero-ablation, we observe a monotonic decrease in the relative significance of the candidate circuit as the size of the random circuits increase. In contrast, with STR ablation, its significance remains constant.

These findings suggest that the knockdown metric alone is insufficient to assess circuit quality, particularly when using STR ablation, as it is sensitive to artifacts from the validation dataset.

| Reference circuit | Induction | IOI | G-T | DS |
|---|---|---|---|---|
| $C^r \sim \overline{C^*}$ | ✓ | ✓ | ✓ | ✓ |
| $C^r \sim M$ | ✓ | ✓ | ✓ | × |

Table 2: A (✓) indicates that knocking down $C^*$ is significantly worse than knocking down $C^r$, while (×) means the converse. $C_r$ is the same size as $C^*$ but draws from different reference distribution.

One explanation is that all edges in the circuit are essential, so knocking down any edge impairs the model's task performance. If a random circuit includes the candidate circuit's edges, the effect is similar. To investigate this, we build on the minimality result.

**Minimality.** Recall from Table 1 that the G-T and IOI canonical circuits are not minimal. In Fig. 3, we gradually knock out more edges from the canonical circuit and report the faithfulness of the modified circuit. For G-T, we can remove around 50% of the edges while retaining the same faithfulness. For IOI, we can remove about 20% of the edges. However, we notice that the faithfulness of IOI does not vary monotonically as more edges are knocked out, revealing the complex mechanisms of circuits (e.g., negative mover heads). Although Docstring is minimal, we can still remove a small subset of edges without impacting faithfulness. This is because the reference edges are all approximately zero and the Docstring circuit was discovered in a toy Transformer model.

## 5    Discussion & Limitations

Do existing circuits align with the circuit hypothesis? We develop a suite of idealized and flexible tests to empirically study this question. The results suggest that while existing circuits do not strictly adhere to the idealized hypotheses, they are far from being random subnetworks.

Our tests successfully differentiate circuits by their alignment with the idealized properties, identifying the Induction circuit [Conmy et al., 2023] as the most aligned. We also demonstrate the limitations of existing evaluation criteria, showing that the knockdown effect alone is insufficient to determine circuit quality and that some benchmark circuits are not minimal.

Our tests and empirical studies have several limitations. The idealized tests are stringent, while the flexible tests are sensitive to circuit size measurements and require careful null hypothesis design. For the non-equivalence and non-independence tests, the desired direction is to retain the null, but we did not empirically study the Type II error associated with these tests. Furthermore, the empirical study uses the original experimental setup, whereas existing work and our ablation studies show that circuits are not robust to changes in the experimental setup.

Despite these limitations, we believe the study provides an overview of the extent to which existing circuits align with the idealized properties. We also believe that the tests will aid in developing new circuits, improving existing circuits, and scientifically studying the circuit hypothesis.

## 6    Acknowledgements

C.S, N.B, C.Z., and D.B. were funded by NSF IIS-2127869, NSF DMS-2311108, NSF/DoD PHY-2229929, ONR N00014-17-1-2131, ONR N00014-15-1-2209, the Simons Foundation, and Open Philanthropy. M.M. was funded by NSF 2153083 and NSF 2337529. A.N. was supported by funding from the Eric and Wendy Schmidt Center at the Broad Institute of MIT and Harvard, and the Africk Family Fund.

The authors thank Sebastian Salazar and Eli N. Weinstein for comments on the manuscript and helpful discussion. They also thank the contributors to the Automatic Circuit Discovery codebase [Conmy et al., 2023], which underlies a significant part of this paper's code.

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

# Appendices

## A   Related Work

This work fits into the broader field of explainable AI [Linardatos et al., 2020], with a particular application to mechanistic interpretability.

**Mechanistic interpretability.**  Olah et al. [2020] introduced the notion of a *circuit* inside a neural network: overlapping units within the network that compute features from other features and can theoretically be understood from the weights. Since then, mechanistic interpretability has proposed many circuits, such as in vision models [Cammarata et al., 2021, Mu and Andreas, 2020, Schubert et al., 2021] and language models [Olsson et al., 2022, Wang et al., 2023, Hanna et al., 2023, Lieberum et al., 2023]. The literature has been particularly successful in explaining Transformers that compute simple, algorithmic tasks [Nanda et al., 2023a, Heimersheim and Janiak, 2023, Zhong et al., 2023, Quirke et al., 2023, Stander et al., 2023].

**Evaluating interpretability.**  Evaluating the quality of interpretability results is an open problem. Some researchers focus on evaluating the *faithfulness* [Jacovi and Goldberg, 2020] of explanations: do they accurately represent the reasoning process of the model? Chan et al. [2022], Geiger et al. [2021] introduced similar formalisms for measuring faithfulness based on causality, with [Schwettmann et al., 2023, Lindner et al., 2024, Friedman et al., 2024, Variengien and Winsor, 2023] providing datasets and methods to generate them.

Early on, Doshi-Velez and Kim [2017] distinguished between functional (without humans) and application-grounded evaluation of interpretability methods, arguing that functional metrics are flawed proxies for the usefulness of the interpretation. Several works [Casper et al., 2023, Mills et al., 2023] adopt this approach to evaluating interpretability.

**Relation to IOI (Wang et al. [2023]).**  The approach proposed in this paper is closest to Wang et al. [2023], which presents three criteria – faithfulness, minimality, and completeness – to evaluate the IOI circuit. Faithfulness is a metric, while minimality and completeness involve searching over the space of circuits.

Our idealized criteria align with the spirit of Wang et al. [2023], but the specific tests differ. Wang et al. [2023] reported a faithfulness score of $0.2$ on the circuit, but the significance of this score is unclear without a reference point. Our sufficiency test contextualizes whether the faithfulness score is significant by comparing it to different reference distributions.

Additionally, Wang et al. [2023] use two other criteria, completeness and minimality. Completeness relates to whether there are parts of the circuit that are not included but may still play a role. They evaluate this by checking if the circuit behaves similarly to the model under knockouts. Minimality checks whether, for a node $v$, there exists a subset $K$ such that including $v$ but removing $K$ significantly changes the score. Both tests require an exhaustive enumeration of circuits and don't use any reference distribution to establish significance, which is an important contribution of our work.

**Relation to ACDC ([Conmy et al., 2023]**  Our work is also related to Conmy et al. [2023]. The ACDC evaluation focuses on the accuracy of edge classification within circuits. While they compare the circuits uncovered by the automated method against circuits found in existing works, we evaluate the quality of the circuits presented in these existing works, i.e., what they used as ground truth.

Additionally, the ACDC algorithm uses a threshold parameter $\tau$ to determine the significance of an edge's relevance to the task at hand. They treat this threshold as a parameter in their search algorithm, sweeping through various values. In contrast, our minimality test offers a principled approach to establish a clear criterion for determining the value of the threshold.

# B   Statistical Tests

## B.1   Equivalence Test

The null hypothesis for the equivalence test is defined as

$$H_0 : \left| \mathbb{P}\left( \Delta\left(X, Y\right) > 0 \right) - \frac{1}{2} \right| < \epsilon, \tag{8}$$

where $\epsilon > 0$ is a user-chosen tolerance parameter.

Given that $\mathbb{1}\{\Delta\left(X, Y\right) > 0\}$ is Bernoulli-distributed under the null hypothesis, we use the test statistic

$$t = \left| \frac{1}{n} \sum_i \mathbb{1}\{\Delta\left(x_i, y_i\right) > 0\} - 1/2 \right|, \tag{9}$$

and choose rejection regions of the form $R_\alpha = \{t \geq c(\alpha)\}$, where $c(\alpha)$ is a yet-to-be defined function of $\alpha$ ensuring that $\mathbb{P}(T \in R_\alpha) \leq \alpha$. Intuitively, $c(\alpha)$ increases (or remains constant) as $\alpha$ decreases. Moreover, because $\mathbb{P}(T \in \{t \geq C\}) = 1$ if $C = 0$, and $\mathbb{P}(T \in \{t \geq C\}) \to 0$ as $C \to \infty$, we know it must be possible to construct at least one function $c(\alpha)$ so that the regions $R_\alpha$ satisfy the requirements of a hypothesis test.

Let $\theta = \mathbb{P}\left( \Delta\left(X, Y\right) > 0 \right)$. By the definition of the hypothesis test and the null hypothesis, we require $\mathbb{P}(T \in R_\alpha) \leq \alpha$ for all $\theta \in \left[ \frac{1}{2} - \epsilon, \frac{1}{2} + \epsilon \right]$. However, notice that for a fixed rejection region $R_\alpha$ and any value $\theta' \in \left[ \frac{1}{2} - \epsilon, \frac{1}{2} + \epsilon \right]$,

$$\mathbb{P}(T \in R_\alpha \mid \theta = \theta') \leq \mathbb{P}\left( T \in R_\alpha \;\middle|\; \theta = \frac{1}{2} + \epsilon \right) \tag{10}$$

Hence, if we have a set $R = \{t \geq C\}$, where $C$ is some constant, $R$ is a valid rejection region for any $\alpha$ such that

$$\alpha \geq \mathbb{P}\left( T \in R \;\middle|\; \theta = \frac{1}{2} + \epsilon \right). \tag{11}$$

Now, construct a function $c(\alpha)$ such that $\mathbb{P}(T \in R_\alpha) \leq \alpha$, and ensure that $c(\alpha_p) = t_{obs}$, where

$$\alpha_p = \mathbb{P}\left( T \geq t_{obs} \;\middle|\; \theta = \frac{1}{2} + \epsilon \right), \tag{12}$$

and $c(\alpha) > c(\alpha_p)$ for any $\alpha < \alpha_p$. We can construct one such function, for example, by letting $c(\alpha) = t_{obs}$ for $\alpha > \alpha_p$, which is admitted by Eq. 11, and by choosing valid values for any $\alpha < \alpha_p$, which is feasible because $\mathbb{P}(T \geq C) \to 0$ as $C \to \infty$. Under this setup, the $p$-value is $\alpha_p$. This follows from the fact that $t_{obs} \in R_{\alpha_p} = \{t \geq t_{obs}\}$, but for any $\alpha < \alpha_p$, $t_{obs} \notin R_\alpha$ as $c(\alpha) > c(\alpha_p) = t_{obs}$.

Finally, we can compute the $p$-value analytically by using the Bernoulli distribution for $\mathbb{1}\{\Delta(x_i, y_i) > 0\}$ with parameter $\theta = 1/2 + \epsilon$,

$$\alpha_p = \sum_{\substack{k \in [n] \\ \left| \frac{k}{n} - \frac{1}{2} \right| \geq t_{obs}}} \binom{n}{k} \left( \frac{1}{2} + \epsilon \right)^k \left( 1 - \frac{1}{2} - \epsilon \right)^{n-k}. \tag{13}$$

An important clarification is that we could have chosen the test statistic to be the estimated value of $\theta$ namely, $\sum_i \mathbb{1}\{\Delta(x_i, y_i)\}/n$ and change the rejection region. In the main text we choose to express it this way for clarity of exposition.

## B.2   Quantile Test

We provide the details of the quantile test used for testing sufficiency, partial necessity, and minimality in Algorithm 1. We state it generally but assume that it would be instantiated for each of the above

cases. Throughout, we assume we are interested in a random quantity $Z$ and want to compare it to a target value $Z^*$. We only use $<$ and $>$ for expository purposes.

For sufficiency, the test corresponds to $l(\cdot) = \mathcal{F}_\tau(M, \cdot)$. For partial necessity, it corresponds to $l(C) \mapsto -F_\tau(M, \overline{C})$.

---

**Algorithm 1:** Tail Test

    **Input:** Population distribution $\mathcal{P}_\mathcal{Z}$, target quantity $Z^*$, quantile $q^*$, number of random
           samples $n$, alternative hypothesis direction: $>$ or $<$, comparison direction: $>$ or $<$,
           significance level $\alpha$
    **Output:** The $p$-value and test statistic

1   $t \leftarrow 0$;

2   **for** $i = 1, \ldots, n$ **do**
3       $Z_i \sim \mathcal{P}_\mathcal{Z}$;
4       **if** *comparison direction is* $<$ **then**
5          $t \leftarrow t + \mathbb{1}\left\{Z^* < Z_i\right\}/n$ ;            `// t will be the test statistic`
6       **else**
7          $t \leftarrow t + \mathbb{1}\left\{Z^* > Z_i\right\}/n$;

8   $p$-value $\leftarrow$ Compute the $p$-value with a binomial test with $t$ successes, $n$ trials, probability of
    success $q^*$, and significance level $\alpha$ using the alternative hypothesis direction;

9   **return** *p-value, $t$*

---

## B.3   Independence Test

We provide details for the independence test used for the partial necessity test. To measure the independence between two variables, we use the Hilbert Schmidt Independence Criterion (HSIC) [Gretton et al., 2007]. HSIC is a nonparametric measure of independence between two random variables. It is based on the idea that if two random variables are independent, then the cross-covariance between the two variables should be zero. It accounts for the nonlinear relationship between the two variables by mapping them into a reproducing kernel Hilbert space (RKHS) and computing the cross-covariance in the RKHS.

**Definition 1** (Hilbert-Schmidt Independence Criterion (HSIC)). *Let $K(x, y) = f(\delta(x, y)/\rho)$ denote a kernel function such as the RBF kernel. Let $\rho$ be a positive parameter called the bandwidth. The Hilbert Schimit Independence Norm is defined as the trace of the covariance between $X$ and $Y$ in the kernel space,*

$$\|C(x, y)\|_F^2 = tr[k_{xy}^T k_{xy}]. \tag{14}$$

A higher HSIC value indicates a stronger relationship between the variables.

The permutation test used for the independence test is detailed in Algorithm 2.

---

**Algorithm 2:** Permutation Test

---

    **Input:** Candidate circuit $C^*$, dataset $\mathcal{D}$, score function $s$, bandwidth $\rho$, number of random
          samples $B$
    **Output:** $p$-value

**1**

**2** $s_{\bar{C}^*} \leftarrow [s(\overline{C^*}(x_1), y_1), \ldots, s(\overline{C^*}(x_n), y_n)]$ ;
**3** $s_M \leftarrow [s(M(x_1), y_1), \ldots, s(M(x_n), y_n)]$;
**4** $t_{\text{obs}} \leftarrow \text{HSIC}(s_{\bar{C}^*}, s_M, \rho)$;

**5**

**6** $t \leftarrow 0$;
**7** **for** $j = 1, \ldots, B$ **do**
**8**      $s_M^{(i)} \leftarrow \text{permute}(s_M)$;
**9**      $t^{(i)} \leftarrow \text{HSIC}\left(s_{\bar{C}^*}, s_M^{(i)}, \rho\right)$;
**10**      $t \leftarrow t + \mathbb{1}\left\{t^{(i)} > t_{\text{obs}}\right\}$;
**11** $p$-value $\leftarrow \frac{t}{B}$; ;                                       `// Approximate` $p$-`value`
**12** **return** *p-value*;

---

# C Minimality

We give a graphical example of the minimality test in Fig. 4.

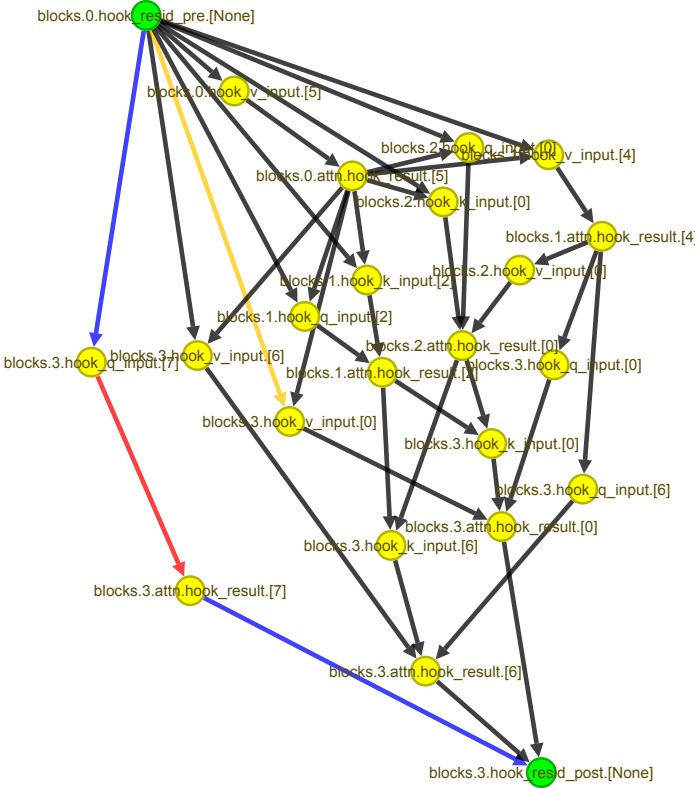

Figure 4: Example of one step of the minimality test for the Docstring task: comparing knocking out a single edge of the candidate circuit (orange edge) against comparing knocking out a random edge of a randomly inflated circuit (the randomly added path is blue, the knocked out edge in the added path is red). Minimality tests whether knocking out the random red edge is more significant than knocking out the orange candidate edge.

# D  Experiment Details

## D.1  Task Description

Below, we provide an extended description of each task and how the experiments were performed.

**Indirect Object Identification (IOI) Wang et al. [2023].**   The objective in IOI is to predict the indirect object in a sentence containing two subjects with an initial dependent clause followed by the main clause. For example, in the sentence "When Mary and John went to store. John gave an apple to" the correct prediction is Mary. We use the dataset provided by Wang et al. [2023] following the structure above. The score used is the logit difference between the correct subject (Mary) and and the incorrect subject (John), and the distribution used to perform STR patching is one where the subjects are replaced for different names, not in the sentence. For example, in the sentence above this could be: "Sarah and Jamie went to the store. April gave an apple to." The candidate circuit we evaluate is the one discovered by Wang et al. [2023] in GPT-2 Radford et al. [2019].

**Induction Elhage et al. [2021].**   The objective for induction is to repeat the completion of a sequence of tokens that previously appeared in the context. For example, in the sentence, "Vernon Dursley and Petunia Durs" the goal is to predict "ley". The score is the log probability assigned to the correct token. We use the dataset provided by Conmy et al. [2023] which contains 40 sequences of 300 tokens from the validation split of OpenWebText Gokaslan and Cohen [2019] filtered to include instances of induction. The circuit we use corresponds to the circuit discovered by Conmy et al. [2023] using zero ablation on a 2-layer 8-head attention-only Transformer trained on OpenWebText. Consequently, we also use zero patching for the experiments with this model.

**Docstring [Heimersheim and Janiak, 2023].**   The objective for the Docstring task is to predict the next variable name inside of a docstring. For example, in

```
def port(self, load, size, files, last):
    """oil column piece
    :param load: crime population
    :param size: unit dark
    :param
```

the model should predict the completion `files`. We use the dataset provided by Heimersheim and Janiak [2023] following the structure above. The STR dataset uses the same input but with the parameter names switched for different ones not in the function. For example, the corrupted input corresponding to the example above replaces `load` by `user` and `size` by `context`. Following Heimersheim and Janiak [2023], for the score we use the logit difference between the correct answer and the most positive logit over the set of alternative arguments, including the ones used for the corrupted example. The circuit we use corresponds to the one provided by Heimersheim and Janiak [2023] which is specified over a 4-layer attention-only Transformer trained on natural language and Python code.

**Greater-Than [Hanna et al., 2023]**   The greater-than task requires performing the greater operation as it appears in natural language. For example, it asks that sentences such as "The demonstrations lasted from the year 1289 to the year 12", are completed with tokens representing two-digit numbers between "89"and "99". Following Hanna et al. [2023], we use as the score function the difference in probability between the two-digit tokens satisfying the relation and those that don't. For STR patching, we use a corrupted datapoint, which replaces the last two digits of the first year with "01". In the example above, this would imply changing "1289" to "1201". The circuit we evaluate for this task is the GPT-2 subgraph provided by Conmy et al. [2023] as a simplification to the original provided by Hanna et al. [2023].

**Tracr [Lindner et al., 2024].**   `Tracr` is a compiler for RASP Weiss et al. [2021], a simple language expressing a computational model for Transformers. We use `Tracr` as by design it provides us with a "ground truth" circuit which allows us to verify the performance of our method. We study two tasks. The first task `tracr-r`, consists of reversing a small sequence of tokens. The second tasks `tracr-p` consists in computing at each position the proportion of tokens corresponding to x that

have been observed. The sequences `<bos> 1 2 3` and `<bos> x a c x` respectively correspond to possible input sequences for the tasks. The sequences `<bos> 3 2 1` and `<bos> 1.0 0.5 0.3 0.5` correspond to the desired outputs respectively. Following Conmy et al. [2023], the score used for both tasks is the sum of token-level $\ell^2$ distances between the desired and produced outputs. For the evaluation dataset we use all permutations of the sequence `1 2 3` for `tracr-r`, and $50$ random 4-character-long sequences consisting of characters in $\{x, a, c\}$ for `tracr-p`. We use zero ablation for both tasks.

### D.2 Software

As part of the paper, we created the *circuitry* package as a wrapper around the *TransformerLens* library, which abstracts away lower-level manipulations of hooks and activations. For a given model, the user specifies a circuit as a subset of nodes and edges and selects an ablation strategy and dataset. The user can then evaluate model performance with respect to the circuit. Our package is implemented efficiently, and can evaluate hundreds of circuits in a few minutes on a single A5000 GPU. The software is available at `https://github.com/blei-lab/circuitry`.

# E    Additional Results

## E.1    Equivalence

The equivalence test evaluates whether the candidate circuit outperforms the original model at least half of the time. As shown in Table 1, none of the natural circuits passed the equivalence test. Table 3 show the test statistics – the proportion of inputs where the candidate circuit outperforms the original model – of all tasks. All circuits except IOI are much worse than the original model at the task. This may be because circuits are only a small proportion of the original model. We omit the `Tracr`-based tasks because their performance is identical to the original model by design (they are ground truth circuits). Thus, in their case, although the null hypothesis is true, the sign test can't be applied.

| Tracr-P | Tracr-R | Induction | IOI | G-T | DS |
|---------|---------|-----------|-----|-----|----|
| – | – | 0.03 | 0.24 | 0.07 | 0.1 |

Table 3: The proportion of times $C^*$ outperforms $M$ on the task. Results for `Tracr`-based tasks are omitted as the performance of the circuit is the same as the original model.

## E.2    Independence

For the independence test, we consider retaining the null as passing the test. As shown in Table 1, the only natural circuit that pass the independence test is induction heads, `Tracr`, the ground truth, circuit does. Table 4 reports the results.

|  | G-T | Induction | IOI | DS | Tracr-P | Tracr-R |
|--|-----|-----------|-----|-----|---------|---------|
| HSIC | 0.000 | 0.001 | 0.001 | 0.001 | 0.000 | 0.000 |
| $p$-value | 0.000 | 0.669 | 0.01 | 0.01 | 1.000 | 1.000 |

Table 4: The HSIC and $p$-value of the independence test.

## E.3    Minimality

To produce the results in Table 1, we set $q^* = 0.9$ and if there exist any edges deemed insignificant, we reject the null hypothesis that the candidate circuit is minimal. We find that only the Induction and `Tracr` circuits pass the minimality test.

In Fig. 5, we plot the scores for knocking out each edge for each circuit. As the `Tracr` circuits are ground truth circuits, all edges are significant relative to the reference distribution.

For the Induction circuit, all edges are also significant relative to the reference distribution. However, for the other circuits, we find that a significant portion of the edges are insignificant. This is especially prevalent for the DS circuit, where less than half of the edges are significantly different from the reference distribution. This suggests that other than the Induction circuit, these circuits are not minimal. Unsurprisingly, for the IOI circuit, we see a few edges that can be removed with little impact to performance, in agreement with Wang et al. [2023].

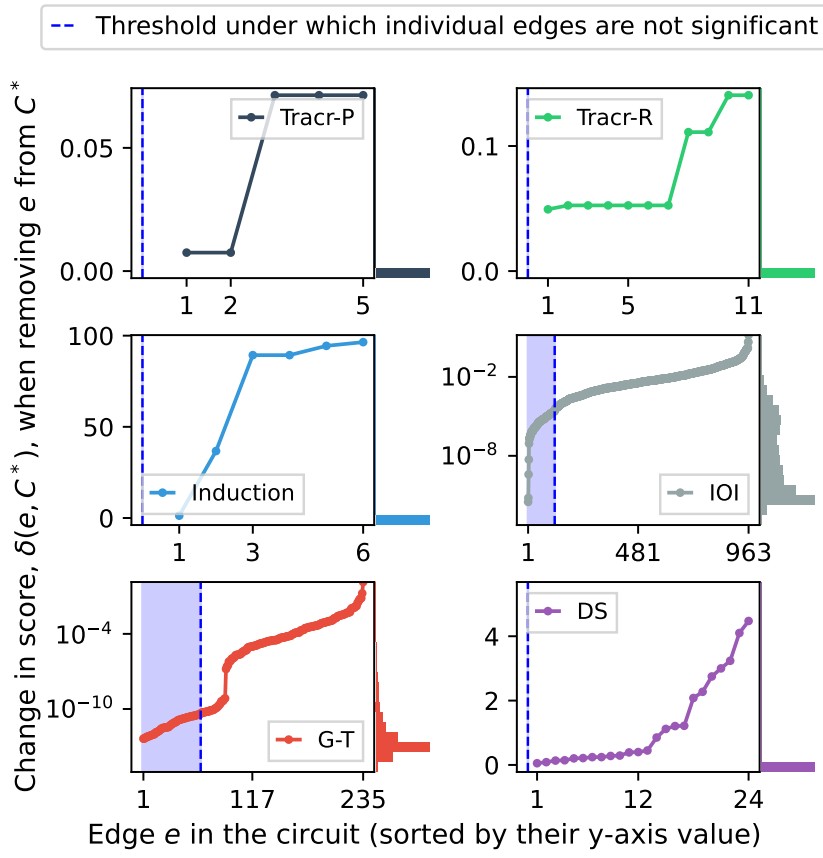

Figure 5: The main figures display the change in task performance score induced by knocking out edge $e$, for every $e$ in each circuit. The changes in score are sorted from low to high along the x-axis. The right-adjacent vertical histograms show the change in task performance scores of the reference edges (ranging from $1000$ to $10,000$ edges). The shaded region covers the individual edges with corrected $p$-values that are below the significance threshold.

# F   Impact Statement

We present a suite of statistical tests to assess whether existing circuits align with the idealized version of circuits. When utilized appropriately, these tests can help identify new circuits, improve existing circuits, and compare the quality across circuits. Thus, we expect these tests to improve the quality of circuits reported in the mechanistic interpretability literature, making them more aligned with the idealized criteria.

We anticipate that the overall effect of this work will be to accelerate progress in mechanistic interpretability and consequently improve our understanding of how LLMs work. This should facilitate explaining and steering model behavior, and possibly "debugging" learned models. At the same time, an improved understanding of model internals may enhance architectures and accelerate capabilities. Additionally, it can open the door to more sophisticated attacks and defenses for various threat models.

It is important to note that our methodology is based on a hypothesis testing framework. Similar to other hypothesis-based tests, there is a potential for misuse or engagement in practices such as $p$-hacking by practitioners. Misapplication of these tests can lead to misleading assurances of robustness that the circuits might not genuinely possess. Furthermore, we assume a fixed experimental setup rather than considering generalization across different setups. The inferences we draw across experimental setups can differ significantly.

If these tests are used to check mechanistic interpretability results for an application of AI, they may give users or developers a misplaced sense of confidence in a faulty hypothesis about neural network internals. However, we believe that this is already a danger with present results, and our work is an improvement in this regard.

