# OpenReview forum: "Hypothesis Testing the Circuit Hypothesis in LLMs"
_NeurIPS.cc/2024/Conference — NeurIPS 2024 poster_

### Official Review · Reviewer_6etf · 2024-06-28

**Soundness:** 3
**Presentation:** 4
**Contribution:** 4
**Rating:** 7
**Confidence:** 4

**Summary:**

This paper considers the problem of formalizing and evaluating the circuit hypothesis. This hypothesis posits that specific subnetworks within a Transformer model are responsible for specific model behaviors. Although there are multiple examples of such circuits that have been manually discovered in the literature, the question of how to precisely judge whether a discovered circuit is indeed responsible for a specific behavior remains open. While past works have proposed various ad-hoc approaches, this paper first presents three criteria that an ideal circuit should satisfy (inspired by past works) along with corresponding statistical hypothesis tests for judging if a given circuit indeed satisfies these criteria. The paper then further presents weaker versions of the hypothesis tests for two out of the three criteria that are more likely to be satisfiable by circuits discovered in practice. These hypothesis tests are then applied to six different circuits that have been proposed in the literature to evaluate the extent to which they satisfy these criteria.

**Strengths:**

This is a nice paper. It is well-written and addresses a question of growing importance. Mechanistic interpretability has emerged as a promising approach for understanding the behavior of trained LLMs. However, given the infancy of the area, the definitions and criteria for evaluating the quality of a mechanistic interpretation are yet to be standardized. This paper fills this need, particularly in the context of circuit analysis. The three proposed criteria of preservation, localization, and minimality make intuitive sense. The experiments with the synthetic circuits also provide evidence in favor of these criteria. I particularly like the proposed use of hypothesis tests to evaluate these criteria. These tests provide an operational definition of an ideal circuit and can be feasibly used in practice for judging new circuits. Overall, I think this paper helps bring some formal discipline to the topic of circuit analysis.

**Weaknesses:**

I have a couple of concerns.

First, the hypothesis test for judging equivalence (Equation 2) feels somewhat unintuitive. Consider the scenario where the circuit C* significantly outperforms the model M on half the inputs and vice versa. Intuitively, one would not consider such a circuit equivalent to the model, yet the hypothesis test would conclude otherwise. Why not define equivalence simply in terms of faithfulness of the circuit C* to model M?

My second concern is the possibility that the three proposed criteria are insufficient for characterizing an ideal circuit, i.e., there might be additional criteria that are needed. However, given that circuit analysis is in its early days, I think the paper already makes useful contributions.

I also have a number of minor comments and questions that I list below:
1. Line 180: Why use ; instead of , in s(C*(x);y)?
2. Equation 2: I think there is a missing abs operation. The definition of the null hypothesis in Appendix B.1 further suggests this.
3. Equation 2,3: Notationally, are lower case and upper case letters being used to represent separate things? For instance, x,y vs X,Y? Why not use lower case x and y in these equations?
4. Equation 5: In practice, there will be randomness also due to the fact that \delta(e,C) is calculated empirically. Does the test account for this randomness?
5. Line 267: The notion of a "reference distribution over circuits from the complement distribution" is not very clear at this stage. I understand what it means based on reading the rest of the paper.
6. Figure 3: The results for the IoITask are a little mysterious. How is the faithfulness of a circuit with some edges removed worse than an empty circuit? Further discussion would be helpful here.
7. Is Algorithm 2 an original contribution or was it presented in Gretton et al., 2007? Some more explanation and intuition about the algorithm would help the reader.
8. It would help to have a more careful comparison with the three criteria for circuits proposed by Wang et al, 2023.

**Questions:**

Please address the questions listed in the previous section.

**Limitations:**

Yes, the paper discusses the limitations and the potential impacts of the proposed work.

---

> ### Author Rebuttal · Authors · 2024-08-06
>
> ## Weaknesses
>
> **First, the hypothesis test for judging equivalence (Equation 2) feels somewhat unintuitive. Consider the scenario where the circuit $C^*$ significantly outperforms the model $M$ on half the inputs and vice versa. Intuitively, one would not consider such a circuit equivalent to the model, yet the hypothesis test would conclude otherwise. Why not define equivalence simply in terms of faithfulness of the circuit $C^{*}$ to model $M$?**
>
> Thank you for raising this great point!
>
> We agree that the test above does not handle this case. However, we intend our suite of tests to be used jointly. The situation you describe would be adequately handled by the sufficiency test, as the circuit you described would surely be unfaithful.
>
> However, we think this test complements a test that only looks at faithfulness for the following reasons:
>
> 1. It can distinguish situations where there is no bias (equally outperform each other) from situations where there is, even when the model is very faithful.
> 2. It is lenient to added variance, which could happen, for example, because we are using patching.
>
> We believe these properties make the test a valuable addition to the suite, as it helps to clarify precisely how the circuits are close or not to the model.
>
>
>
> **My second concern is the possibility that the three proposed criteria are insufficient for characterizing an ideal circuit, i.e., there might be additional criteria that are needed. However, given that circuit analysis is in its early days, I think the paper already makes useful contributions.**
>
> We agree! We hope future work builds on these criteria.
>
> ## Minor comments and questions
>
> Thank you so much for the careful read! Your feedback is greatly appreciated.
>
> - **Line 180: Why use ; instead of , in $s(C^{*}(x);y)$?**
>
>     That’s a great question! This is a typo --- it came about because $y$ can be a parameter of the task, rather than the classic label. For example, it can be the index of the subject and object in IOI. We spent a fair bit of time debating whether to treat that as a parameter with `;` or a variable which we denote with `,`. However, we realize we have been inconsistent with the notation and have changed all of them to `,`.
>
> - **Equation 2: I think there is a missing abs operation.**
>
>     Yes. Apologies about that.
>
> - **Equation 2,3: Notationally, are lower case and upper case letters being used to represent separate things?**
>
>     It is a typo. We have changed them to lowercase.
>
> - **Equation 5: In practice, there will be randomness also due to the fact that $\delta(e,C)$ is calculated empirically. Does the test account for this randomness?**
>
>      This is a great question! Randomness in a dataset would definitely introduce additional variance. We sidestep this issue by defining the task with respect to a fixed dataset. But a different test could be designed to account for a finite dataset. In the flexible tests, we only account for the randomness arising from the sampled circuits and edges.
>
> - **Line 267: The notion of a "reference distribution over circuits from the complement distribution" is not very clear at this stage.**
>
>     Thank you for pointing this out! We’ve rephrased it as ''reference distribution over circuits from the complement distribution, i.e., circuits that do not overlap with the candidate circuit.'' Is this clearer?
>
> - **Figure 3: The results for the IoITask are a little mysterious.**
>
>     We believe this is because of the ''Negative Name Mover Heads'' discovered in the original IOI paper [1], which write in the opposite direction of the name mover heads.
>
> - **Is Algorithm 2 an original contribution or was it presented in Gretton et al., 2007?**
>
>     Algorithm 2 has two components, the permutation test and HSIC. The permutation test is a classic hypothesis test by Fisher [2] and Pitman [3]. It is a nonparametric test to aim to show whether the observed statistics could be drawn from the “permuted distribution”. In our setup, we randomize the complement circuit from the observed y -- creating an independent null. If the observed statistics fall outside of the independent null, then we reject the hypothesis. The HSIC is proposed measures in Gretton et al [4] to measure the independence between two variables in the kernel space.
>
> - **It would help to have a more careful comparison with the three criteria for circuits proposed by Wang et al, 2023.**
>
>     Yes, we absolutely agree. It is discussed in the extended related work in line 486, Appendix A.
>
> ## References
> [1] Kevin Wang, Alexandre Variengien, Arthur Conmy, Buck Shlegeris, & Jacob Steinhardt. (2022). Interpretability in the Wild: a Circuit for Indirect Object Identification in GPT-2 small.
> [2] Fisher, R. A. (1935). The design of experiments. New York, NY: Hafner.
> [3] Pitman, E. J. G. (1937). Significance tests which may be applied to samples from any population. Journal of the Royal Statistical Society. Supplement, 4, 119-130, 225-232.
> [4]  Gretton, A., Fukumizu, K., Teo, C., Song, L., Schölkopf, B., & Smola, A. (2007). A Kernel Statistical Test of Independence. In Advances in Neural Information Processing Systems. Curran Associates, Inc.

---

> > ### Comment · Reviewer_6etf · 2024-08-09
> >
> > Thank you for the detailed response! I will keep my score. It will be helpful to include the discussion of Equation 2 in the paper.

---

> > > ### Author Response · Authors · 2024-08-12
> > > **Thank you!**
> > >
> > > Thank you for responding to our rebuttal and for your support of the paper! We will definitely include the discussion of equation 2 in the paper.

---

### Official Review · Reviewer_NNjL · 2024-07-11

**Soundness:** 3
**Presentation:** 3
**Contribution:** 3
**Rating:** 6
**Confidence:** 3

**Summary:**

This paper proposes a set of tests to evaluate how well a "circuit" meets its desired properties.
- Here a circuit refers to a subnetwork, which could be either synthetic (e.g. constructed according to RASP) or discovered in a trained model.
- The desired properties considered in this paper are:
  - *faithfulness*, where the circuit should preserve the performance of the original model;
  - *localization*, where removal of the circuit should alter the model's output;
  - *minimality*, which says the edges in a circuit (treated as a computational graph) should not be redundant.

There are 3 "idealized tests" intended as necessary (but not sufficient) conditions:
- **Equivalence**, where the null hypothesis is that the circuit $C^*$ preserves the performance of the original model $M$.
  - The test checks whether $C^*$ and $M$ have an equal chance of outperforming each other, which is a necessary condition (but not sufficient): rejecting the null means that $C^*$ necessarily does not preserve the performance.
- **Independence**, where the null hypothesis is that the output of the model after removing $C^*$ is independent of the output of the original model.
  - The test computes the HSIC between the performance of $C^*$ and $M$.
  - Note that this is a stringent requirement; this will later be modified into a "flexible" test.
- **Minimality**, where the null hypothesis is that all edges in $C^*$ are necessary and not redundant.
  - The test checks whether the amount of performance change by removing an edge in $C^*$ is more than the amount of change caused by removing an edge that is believed to be redundant.

The paper also proposes 2 "flexible tests", where the difficulty of the test can be gradually varied by choosing different reference distributions when computing the test.
- **Sufficiency**, which computes the probability (over random circuits in a reference distribution) that $C^*$ is more faithful than a random circuit.
- **Partial necessity**, which checks whether with high probability, removing $C^*$ leads to a worse performance than removing a random circuit.

The paper then applies these tests on 2 synthetic circuits and 4 circuits manually discovered from trained Transformers.
- Synthetic circuits: for 2 types of the Tracr task, where the circuits are given as RASP programs.
- Discovered circuits: Indirect object identification (IOI), Induction, Docstring (DS), Greater-Than (G-T).

It finds that the synthetic circuits pass all 3 tests.
In contrast, the discovered circuits fail the test to various extents, but are still far from random.
Moreover, the flexible tests provide a more fine-grained understanding than the idealized tests (with binary outcomes).

**Strengths:**

- The paper provides a quantitative way to evaluate the concept of "circuit" in models, by adopting the hypothesis testing framework. The proposed tests correspond to desired properties a circuit.
- The proposed tests are more thorough and fine-grained than some existing evaluation criteria, such as the knockdown effect.
- The paper are clearly written.

**Weaknesses:**

- I'm concerned about the practical applicability of the tests.
As the paper also mentioned, the proposed tests are either too stringent, or can be sensitive to the circuit size and the choice of the reference distribution.
- The implications and use cases of the tests could be better discussed; e.g. please see questions below.

**Questions:**

- About independency: neural networks are often overparameterized and hence likely contain a high level of redundancy. What if there are multiple circuits that are each faithful and minimal but are similar to each other? These seem to me should be considered as valid circuits, but they would violate independency.
- About minimality, I don't see why the randomly added edge would lead to a small $\delta(e^I, C^I)$: it's possible that adding a random connection would change the performance non-trivially (but likely negatively). For example, even a simple residual term would change the scale of the output and affect subsequent computations.
- Could you comment on the effect of different choices of activation patching?
- Could you comment on how the proposed tests can inform interventions on training (e.g. as regularization terms)? The hope is that this could make the trained networks more likely to contain circuits that more closely satisfy the desiderata.
- Can the proposed tests help with coming up a more precise definition of circuits, or inform how we should choose the granularity of the definition of "nodes"?

Minor clarifications:
- Eq (2): should the LHS be taking an absolute value?
- Line 246: not sure I understand what “a lucky draw” means; e.g. it could mean a draw that is better than q* fraction of random circuits?
- Line 250: what does "supersets of C" and "comparable to C" mean?

**Limitations:**

The paper discusses the technical limitations.

There is no direct societal implication.

---

> ### Author Rebuttal · Authors · 2024-08-06
>
> ## Weaknesses
>
> **I'm concerned about the practical applicability of the tests (...) can be sensitive to the choice of the reference distribution.**
>
> Thank you for raising this important point! We believe that the main objections stem from a small but important misunderstanding of the paper's main goal.
>
> The review states:
> > This paper proposes a set of tests to evaluate how well a "circuit" meets its desired properties.
>
> We politely disagree with this goal. Our goal is to formalize the implications of the circuit hypothesis---"LLMs implement tasks through circuits." We demonstrate consistency with synthetic circuits, and we then study how existing circuits align with these idealized properties. Our goal is not to create a checklist that evaluates which circuits are better or worse.
>
> This subtle but important difference leads to different interpretations of the results. If a circuit does not align with the circuit hypothesis, it does not mean the circuit is "undesirable"; rather, it could indicate that the idealized version of circuits differs from how neural networks actually encode tasks. Our paper aims to provide tools to quantify the extent to which the idealized circuit hypothesis aligns with circuits in practice.
>
> Thus, the practical applicability of our tests is to provide a nuanced understanding of the alignment between the circuit hypothesis and discovered circuits. Currently, circuits found in transformers rarely pass the stringent tests due to issues like the redundancy mechanism, as the reviewer correctly points out. However, future models could exhibit more modular and circuit-like behavior. Our tests can help determine when this happens.
>
> ## Questions
>
> **What if there are multiple circuits that are each faithful and minimal, but are similar to each other?**
>
> We agree that neural networks implement redundancy, making the independence test difficult to pass. This suggests that the network is not implementing the tasks similarly to how the idealized version of  "circuits:' and neural networks behave. Our tests help illustrate how the neural network in practice does not align with the idealized circuit hypothesis.
>
> **About minimality, I don't see why the randomly added edge would lead to a small $𝛿(𝑒^𝐼,𝐶^𝐼)$**
>
> Thank you for the question. A key component of the circuit hypothesis is that edges not part of the circuit can be removed without significantly affecting model performance, hence our design of the minimality test. While "removing" is a common term, this can be more complex than simply setting values to zero, as is the case with STR patching [1]. This is detailed in lines 109-124 of the paper. This ablation method is chosen to maintain the edge magnitude and avoid the issues you described. Further clarifications on activation patching are provided in the question below.
>
> **Could you comment on the effect of different choices of activation patching**
>
> Congruent with existing findings [1], we found that the circuit is indeed sensitive to the ablation scheme that was used to “discover” it. This is illustrated in Figure 6 of the paper. We present the results with the original method used to discover the circuit with the aim of fairness. Appendix D.1 clarifies which ablation methods were used with which dataset.
>
> **Could you comment on how the proposed tests can inform interventions on training**
>
> Thank you for the question! These tests can indicate if certain training methods or architectural variations are more likely to produce idealized circuits. For instance, if circuits in a transformer trained with a specific regularizer align more with the tests, this suggests that this method is more effective at producing circuits.
>
> However, it is unclear how to incorporate these tests during training. These tests are designed for specific "hand-crafted" tasks $\tau$, while pre-trained models are not trained this way. Additionally, systematizing the creation of many such $\tau$ or deriving a differentiable loss from the tests remains a challenge.
>
> **Can the proposed tests help with coming up a more precise definition of circuits?**
>
> Thank you for the question! It depends on what you mean by this.
>
> When we consider the Transformer and related architectures as computation graphs, any meaningful definition of a circuit is likely to be similar to the one we have provided. This similarity arises because the computation graph framework inherently constrains how circuits can be defined. Consequently, it would be challenging to find a more precise definition of a circuit within this framework without it closely resembling our existing definition.
>
> However, if you are asking about the level of granularity at which we apply the definition (e.g., interventions at a node level vs. edge level), then we think our tests would be useful. This would certainly be an interesting experiment to explore in future work.
>
> ## Minor Clarifications
>
> - **Eq (2): should the LHS be taking an absolute value?**
>     Yes! Thank you for pointing it out!
>
> - **Line 246: not sure I understand what “a lucky draw” means**
> We mean the following:
>     If with a small probability (e.g., 10%), we could have randomly drawn a circuit just as faithful as the candidate circuit, then the candidate circuit is simply a "lucky draw" from the reference distribution. If the candidate circuit is significantly more faithful than the 90% most faithful random circuit, then it is not just a lucky draw.
>
> - **Line 250: what does "supersets of C" and "comparable to C" mean?**
> A circuit is defined as a set of nodes and edges. A superset of C is a circuit that contains all edges and nodes in C. For a circuit to be "comparable to C" means that it will have similar faithfulness to C. We expect a superset of C to be comparable to C.
>
> ## References
> [1] Fred Zhang, & Neel Nanda. (2024). Towards Best Practices of Activation Patching in Language Models: Metrics and Methods.

---

> > ### Author Response · Authors · 2024-08-12
> >
> > Dear Reviewer, thank you again for the time and effort you've dedicated to reviewing our work.
> >
> >  We believe our responses address the concerns raised in your reviews. As the discussion period is nearing its conclusion, if you find that any aspects of our responses require further clarification or discussion, we are eager to engage in constructive dialogue!

---

> ### Comment · Reviewer_NNjL · 2024-08-13
>
> Sorry for the delay in my response, and thank you for the clarifications!
>
> My main concerns have been addressed, and I've raised my score. I'd appreciate more clearly stating the paper's goal in the camera ready.
>
> Another question please: the word "circuit" has different meanings with different implications on generalization. For instance, circuit in the complexity sense (e.g. a boolean circuit) refers to a unit with well-defined computation, and hence would have perfect OOD generalization (where OOD refers to a change of distribution over the inputs). In contrast, there's typically no generalization guarantee when referring to circuits as subnetworks. Could you share your thoughts on what implications on generalization could we get by interpreting a network through its subnetworks?

---

> > ### Author Response · Authors · 2024-08-14
> > **Thank you for the response and the question!**
> >
> > Thank you for your response! We are glad that we have addressed your concerns. We will ensure that the paper's goal is clearly stated in the camera-ready version.
> >
> > Regarding the question about circuit generalization, this is an excellent point and aligns with our current research. Currently, a circuit is defined with respect to a dataset, and circuits do not generalize well if the dataset is significantly changed, even if the underlying task remains the same. One possible reason is that multiple circuits can replicate the model behavior for a given dataset, but only a few may generalize across different datasets. For example, a circuit for mathematical computation can perform the Greater Than task, as can a circuit that can output a larger number after some lower number.
> >
> > We believe finding a circuit that generalizes across different datasets of the same tasks bring us closer to the "circuits" in a complexity-theoretic sense.

---

### Official Review · Reviewer_wEmx · 2024-07-12

**Soundness:** 2
**Presentation:** 4
**Contribution:** 2
**Rating:** 6
**Confidence:** 3

**Summary:**

The paper operates in the framework of mechanistic interpretability of transformer models, where it is assumed that subgraphs (i.e. circuits) of the computational graph determined by the model implement specific capabilities of the latter. It defines a set of properties that an ideal circuit should have, namely equivalence, independence and minimality. Afterwards, it proposes two sets of hypothesis tests. The first set identifies a more strict hypothesis-testing framework to determine if a circuit satisfies these properties. The second proposes some more flexible tests for the first two properties of ideal circuits. Afterwards, these hypothesis tests are used on synthetic and manually discovered circuits in the literature on GPT-2 small and other small transformers.

**Strengths:**

- Overall very well-explained paper, particularly the part on mechanistic interpretation.
- Crafting of the hypotheses for testing is original and well thought. It follows logically from the properties defined and is well explained.
- Different types of tests were performed with different granularity.

**Weaknesses:**

- I believe it would have been nice to perform further tests to understand whether the idealized tests have some applicability on discovered circuits (or if they always lead to the null being rejected), potentially by also altering the discovered ones.
- Would have found it interesting to propose some ideas on how to use these tests for circuit discovery.

**Questions:**

- I believe there is a typo in the description of Table 1 on page 7, where it is claimed that “A (✓) indicates the null hypothesis is rejected”, while I believe that this means that the circuit “passed the test” so that the null-hypothesis is not rejected.
- When referring to the Bonferroni correction, was the base one used or the Bonferroni-Holm one? The latter is always more appropriate as it is less conservative and leads to a more powerful test.

**Limitations:**

-

---

> ### Author Rebuttal · Authors · 2024-08-06
>
> ## Weaknesses
> **I believe it would have been nice to perform further tests to understand whether the idealized tests have some applicability on discovered circuits. (or if they always lead to the null being rejected), potentially by also altering the discovered ones.**
>
> **Would have found it interesting to propose some ideas on how to use these tests for circuit discovery.**
>
>
> Thank you for bringing this up! We agree that these tests can be applied to improve circuit discovery algorithms, such as ACDC [1].
>
> For example, it would be interesting to apply the minimality test for edge pruning in Algorithm 1 of ACDC [1]. In ACDC, the edges are pruned by checking that removing an edge does not increase the KL divergence between the outputs and the current circuit by more than a threshold, $\tau$. This threshold $\tau$ is treated as a hyperparameter. In contrast, using the minimality test, we can determine the importance threshold in a principled way.
>
>
>
> However, we omitted further discussion of these ideas because they are not the main focus of this paper, which is to formalize the circuit hypothesis, develop appropriate tests, and apply them to study the extent to which the circuit hypothesis holds. Nevertheless, we believe that applying our proposed tests in novel circuit discovery algorithms is an exciting area for future research.
>
>
>
> ## Questions
>
> **I believe there is a typo in the description of Table 1 on page 7**
>
> Thank you for pointing out the typo; you are absolutely right. We have updated the paper to reflect that.
>
> **When referring to the Bonferroni correction, was the base one used or the Bonferroni-Holm one?**
>
> Thank you for the great suggestion. We used the Bonferroni correction in the paper because it is easier to explain, but you are right that Bonferroni-Holm is uniformly more powerful. We have made a note about it in the paper and will incorporate it in the paper's code package. In our experiments, we explored another less conservative method, the Benjamini–Hochberg method, but we did not observe a difference in the main findings. So we opted for Bonferroni correction for simplicity.
>
> ## References:
>
> [1]  Arthur Conmy, Augustine N. Mavor-Parker, Aengus Lynch, Stefan Heimersheim, & Adrià Garriga-Alonso. (2023). Towards Automated Circuit Discovery for Mechanistic Interpretability.

---

> > ### Comment · Reviewer_wEmx · 2024-08-09
> >
> > Thanks for your answer. I will keep my score.

---

> ### Author Response · Authors · 2024-08-12
>
> We thank the reviewer for their comments , response, and support of the paper!

---

### Author Rebuttal · Authors · 2024-08-06

We thank the reviewers for their thoughtful reviews and support of the paper.

We are pleased to see that the reviewers find the paper well-explained, with original and well-considered hypothesis tests (Reviewer wEmx); that it is clearly written, offers a quantitative approach to evaluating the concept of ''circuit'' in models, and that proposed tests are more thorough and fine-grained than some existing evaluation criteria (Reviewer NNjL); and that it addresses a question of growing importance, bringing formal discipline to the topic of circuit analysis (Reviewer 6etf).

We address the questions individually below.

---

### Decision · Program_Chairs · 2024-09-25

**Decision:**

Accept (poster)

**Comment:**

This paper describes a novel approach for testing the hypothesis that specific functions in LLMs are carried by small subnetworks (i.e., circuits).

There was unanimous agreement among the reviewers that the approach is novel and interesting.
Some concerns were raised including the practical applicability of the approach but the rebuttal appears to have addressed all the reviewers' concerns.

All in all there appears to be sufficient support from the reviewers in favor of accepting this paper for it introduces a novel formalization and evaluation method for the circuit hypothesis in LLMs. Hence, the AC recommends the paper to be accepted.